# Perceptions of and Preparedness for the Application of Pharmacoeconomics in Practice, among Final Year Bachelor of Pharmacy Students in South Africa: A National Cross-Sectional Study

**DOI:** 10.3390/pharmacy11020054

**Published:** 2023-03-14

**Authors:** Carlien Schmidt, Moliehi Matlala, Brian Godman, Amanj Kurdi, Johanna C. Meyer

**Affiliations:** 1Department of Public Health Pharmacy and Management, School of Pharmacy, Sefako Makgatho Health Sciences University, Molotlegi Street, Garankuwa, Pretoria 0208, South Africa; 2Department of Pharmacoepidemiology, Strathclyde Institute of Pharmacy and Biomedical Sciences, University of Strathclyde, Glasgow G4 0RE, UK; 3Centre of Medical and Bio-Allied Health Sciences Research, Ajman University, Ajman 346, United Arab Emirates; 4Department of Pharmacology, College of Pharmacy, Hawler Medical University, Erbil 44001, Iraq

**Keywords:** pharmacoeconomics, universal health coverage, BPharm final year students, cost effectiveness, South Africa

## Abstract

For the improvement of access to health, many countries including South Africa, have adopted universal healthcare. However, this requires skills to apply health technology assessments for the facilitation of investment decisions. This study aimed to ascertain final year Bachelor of Pharmacy (BPharm) students’ perceptions of the relevance of pharmacoeconomics in pharmacy practice, and their level of preparedness to apply pharmacoeconomic principles, using a quantitative, cross-sectional, and descriptive design. Data were collected using a self-administered questionnaire over 12 months, and included student demographics, knowledge about pharmacoeconomics and its applicability in practice, as well as students’ satisfaction with the appropriateness of the curriculum content. Five of nine universities offering pharmacy education took part. The overallstudent response rate was 38.1% (189/496), with 26.2% (45/172) of students signifying a good understanding of basic pharmacoeconomic concepts. Pharmacoeconomics application in South Africa was perceived to be relevant by 87.5% (140/160); however, 47.0% (79/168) felt they were not prepared to apply pharmacoeconomic principles in medicine management, and 86.7% (137/158) wanted to acquire additional pharmacoeconomic knowledge. Whilst students’ perceptions of the relevance of pharmacoeconomics were positive, results indicated a gap in knowledge, understanding, and application. Addressing this gap may increase students’ preparedness to apply pharmacoeconomic principles and better equip them for the practical application of pharmacoeconomics post qualification. Consequently, we have started this process.

## 1. Introduction

Pharmaceutical therapy-related expenditure has become an essential consideration to healthcare payers worldwide focusing on pharmacoeconomic analyses, with medicine expenditure in some low-and middle-income countries (LMICs) accounting for up to 70% of total healthcare expenditure [1,2,3,4,5]. In high-income countries, there is an increasing focus on new medicines for cancer and orphan diseases as requested prices increase with often limited health gain coupled with the potential to overwhelm universal healthcare systems with growing expenditures [6,7,8]. These concerns have intensified the focus on the necessity for the scientific valuation of costs and consequences of pharmaceutical treatments, including vaccines to guide future investments and policy decisions [1,9]. Limited healthcare resources have also increased interest in assessing the value and feasibility of funding competing healthcare treatments and programmes by performing pharmacoeconomic evaluations, especially among LMICs [3,10,11,12,13,14]. The rise in pharmacoeconomic research application is expanding the need for qualified individuals, who are able to analyse and understand research findings and translate these into practice, especially among LMICs with resource and other concerns [3,10,15,16,17,18,19,20,21].

To date, South Africa has utilised dedicated methods in specific situations during reimbursement and pricing decision-making process for medicines rather than a broader use in priority setting, where there are competing demands across disease areas [9,22,23]. Pharmacoeconomic submissions to the South African National Department of Health (NDoH) Pricing Committee have taken place voluntarily and for selected medicines in the private health sector [22,23,24]. This is due to the fact that South Africa currently has an unequal two-tier healthcare system, with a public and private sector. The public sector, which is state funded caters for approximately 80% of the population with the goal of universal healthcare (UHC) [25,26]. The private sector, which caters for approximately 20% of the population, is largely funded through medical aid contributions or health insurance [27]. Consequently, medicines in the public sector are usually subject to tenders as they have typically lost their patents. [23,28]. This is not the case for possible new medicines in the private sector, with pharmacoeconomic guidelines in South Africa initially developed for the private sector, recommending a third-party payer perspective [22]. Nonetheless, pharmacoeconomic analyses are now emerging in the public sector to help appraising different treatment approaches, including different public health approaches, with competing demands for finite resources [29,30,31,32,33,34,35,36,37]. This inconsistent use of pharmacoeconomic submissions may imply that national pharmacoeconomic evaluations and education settings are still emerging, although there are moves to improve submissions through international comparisons [10,23,38]. In this regard, South Africa will soon follow in the footsteps of many international countries concerning pharmacoeconomic research application when the National Health Insurance (NHI) system, aimed at universal health coverage, is fully functional [25,26,38]. In the near future, it is believed that health technology assessment methodologies will be used to prioritise interventions in key areas, including health promotion, disease prevention and treatment, with the most cost-effective, evidence-based therapies and strategies being deployed and paid for under the NHI [25]. This is similar to situations in other LMICs [11,12,13,14]. We will continue to thoroughly monitor the situation with its implications for the necessity to increase pharmacoeconomic understanding among healthcare students, with the likeliness of strengthening the pharmacoeconomic guidelines in South Africa.

With their knowledge of medicines and their costs, pharmacists are uniquely equipped to use pharmacoeconomic analyses to influence expenditure on medicines and the distribution of resources for medicines [1,3,15,23,25,26]. This builds on previous approaches, including ABC- and VEN-analyses of medicine use and expenditures in hospitals [15,39,40,41]. The South African National Drug Policy covers various activities contributing to effective medicines management. Within this policy, the pharmacist’s role is to ensure that the South African population receives the medicines they need at a cost that is affordable to them, and the healthcare system, is also clearly stipulated [24]. This role further enhances the need for appropriate pharmacoeconomic knowledge and skills among pharmacists going forward.

The number of academic institutions providing education on pharmacoeconomic analyses has grown internationally over the past two decades, with many institutions also increasing the extent of health economics coursework amongst undergraduate pharmacy students [3,16,17,18,19,20]. Similarly, education on pharmacoeconomics is now incorporated into the South African Bachelor of Pharmacy (BPharm) programmes, complying with the South African Pharmacy Council’s exit level outcomes for entry-level pharmacists and the qualification standards of the South African Qualifications Authority [21,42]. The education of pharmacoeconomic principles in South Africa was briefly covered in a study published in 2005 [3]. While this study mentioned a requirement for increased education on pharmacoeconomics in developing countries, the individual perceptions of undergraduate pharmacy students in South Africa regarding pharmacoeconomics have yet to be robustly investigated [3]. A South African-based study focusing on pharmacoeconomics in the healthcare system in Gauteng province highlighted that pharmacoeconomics education is essential for increased awareness and understanding of the subject among healthcare decision-makers, with most respondents concurring that pharmacoeconomics education would aid them in their scope of practice [43]. However, this has not been considered further.

A shortfall in healthcare professionals’ and students’ knowledge and understanding of pharmacoeconomics, and its application in medicines management, appears to be universal across countries, including LMICs [16,44,45,46]. The results of a South African study conducted in 2005 highlighted a lack of pharmacoeconomic knowledge among healthcare workers, epidemiologists, and trained staff, subsequently leading to an absence of measures to control resources in both the public and private healthcare sectors [43]. This is starting to be addressed in South Africa and other African countries with groups, such as HTAi having dedicated interest groups for developing countries (https://htai.org/hta-in-developing-countries/; accessed on 24 November 2022) as well as ISPOR with its African chapter (https://www.ispor.org/member-groups/global-groups/networks/africa-network/executive-committee; accessed on 24 November 2022). Applied pharmacoeconomics is often viewed as an important skill set for pharmacists internationally, and is acknowledged for improving health system performance across countries [3,43,45,47,48]. Despite this positive view on the role of pharmacoeconomics in maximising patients’ outcomes from the available healthcare resources, healthcare students and decision-makers commonly feel unprepared to apply pharmacoeconomic principles in practice [16,45,48]. Researchers universally suggest increased education regarding pharmacoeconomics during undergraduate and postgraduate health education programmes to address current gaps in knowledge and the application of pharmacoeconomic analyses [3,10,16,17,43,44,47].

The expected future use of pharmacoeconomic evidence in South Africa as part of the government’s NHI plan clearly indicates that pharmacists will increasingly need knowledge, skills, and capabilities for critical analyses and implementation of pharmacoeconomic research findings. One way to support this is through adequate education in this field among BPharm students in South Africa [3,16,20]. Consequently, this study was undertaken to determine final year BPharm students’ perceptions of the relevance of pharmacoeconomics in their future practice in South Africa and their level of preparedness to apply their knowledge in practice, and be able to undertake and critically review pharmacoeconomic studies in the future.

## 2. Materials and Methods

### 2.1. Study Design and Population

This was a descriptive cross-sectional study among final-year BPharm students enrolled at the nine South African universities offering the BPharm programme. Therefore, all nine universities were invited to allow their final year BPharm students to participate in the study. Specific inclusion criteria for students were: (i) Final year students enrolled in a BPharm programme at a South African university; and (ii) students willing to participate in the study.

### 2.2. Data Collection Instrument and Procedure

Data were collected through a structured, self-administered questionnaire available in English. The questionnaire content was based on a comprehensive review of literature sources, the current pharmacoeconomic curricular content for the South African BPharm programme, and pharmacoeconomic theoretical principles [3,16,17,19,42,44,45,47,49,50,51]. Three pharmacoeconomic experts initially reviewed the questionnaire, with their feedback subsequently incorporated into the revised questionnaire.

A pre-test of the questionnaire was conducted among six pharmacist interns at the SMU School of Pharmacy to determine its face validity, length of completion, and relevance of the questions included.

The final questionnaire consisted of 31 questions (Appendix A), grouped into six sections to collect students’ demographic information, evaluate exposure to training on pharmacoeconomic principles and techniques during BPharm enrolment, and assess perceptions and preparedness pertaining to pharmacoeconomic analyses using a five-point Likert scale [44,45]. A Likert scale was considered the most appropriate measurement scale to assess respondents’ perceptions and preparedness, as it allows for the measurement of different levels of agreement and disagreement. As a result, providing good insights into respondents’ perceptions. Furthermore, the Likert scale has been used extensively in descriptive and quantitative studies across countries [52,53,54].

Students completed a paper-based version of the questionnaire, with completed questionnaires returned to the first author in a sealed envelope, or completed the questionnaire electronically using SurveyMonkey^®^, an online survey platform. Data were collected between November 2018 and December 2019, varying among universities with an average of 3 months per university.

### 2.3. Data Capture and Analysis

Prior to analysis, participating universities were anonymised and recoded as “A”, “B”, “C”, “D”, and “E”. Captured data were proofread, cross-checked, and discrepancies resolved. Data analysis was descriptive and undertaken in custom formulated Microsoft Office Excel^®^ spreadsheets. Categorical variables were summarised by frequency counts and percentages.

Responses to the five-point Likert scale questions were condensed into three categories to facilitate the analysis and interpretation of results. Responses to open-ended questions were typed, categories were manually created, and responses were coded into these categories and counted where applicable.

Even though comparing the individual universities was not the primary aim of this study, a sensitivity analysis was conducted to assess the effect of variation in response rates amongst the universities on the study outcomes. For this purpose, we grouped universities based on their response rates into “Low-“, “Medium-“, and “High-“ response rates. Subsequently, we compared the scores for two randomly selected study outcomes, namely “Level of understanding of pharmacoeconomics” and “Preparedness to apply pharmacoeconomics in practice” between the three groups, using one-way ANOVA and Fisher Exact tests, respectively. Herein, *p*-values < 0.05 were considered as statistically significant.

### 2.4. Ethical Considerations

The Sefako Makgatho University Research Ethics Committee provided ethics clearance for the study (SMUREC/P/97/2018:PG), after which the nine universities were invited to participate in the study. Upon acceptance of the invitation by a university, permission to conduct the study was formally requested, which included submission of the protocol to the respective university’s research ethics committees. Only upon receipt of permission and ethics clearance from a particular university students were invited to participate in the survey. Participation was voluntary, responses were anonymous, and no personal, identifying information was collected. Students participating in the survey first provided informed consent before completing the questionnaire. Data were treated as highly confidential, with completed questionnaires stored under secure conditions. All data are stored securely for future reference and for a period of 5 years, after which it will be destroyed according to university policies.

## 3. Results

### 3.1. Response Rate

Eight of the nine invited universities offering the BPharm programme in South Africa responded positively to the invitation. Five of the eight universities were able to provide ethics clearance for their students to participate in the study during the allocated study period. The ethical clearance process at two of the three remaining universities was delayed considerably, with permission granted only after data collection for the study had been concluded. No further response was received from the one remaining university.

The final target study population from the five universities included 496 final year BPharm students, from which an overall response rate of 38.1% (189/496) was obtained, ranging from 18.1% to 93.9% at individual universities (Table 1). Four of the universities’ students responded using the paper-based version of the questionnaire, while the students at one of the universities responded using the electronic platform (Table 1). Overall, 48.1% (91/189) of students answered all questions in the questionnaire. As a result of inconsistent responses, sample sizes varied between questions. Twenty-two of the 189 students (11.6%) provided additional comments on pharmacoeconomics.

### 3.2. Respondent Demographics

Table 2 demonstrates that the mean age of students was 24.3 years (SD = 2.34), ranging between 20.0 and 32.9 years, with the majority being female (71.4%). A few students (7.1%; 13/184) held other degrees. Nearly half of the students surveyed (49.2%) intended to complete their internships at public sector institutional pharmacies.

### 3.3. Pharmacoeconomics Education during the BPharm Programme

Overall, 37.8% (*n* = 62) of BPharm students in this study indicated that pharmacoeconomics was covered under “Hospital Pharmacy Practice”-related subjects (Table 3). Of the 178 students, 74 (41.6%) indicated that pharmacoeconomics was presented during their fourth year of the programme. The majority of students (87.3%; 151/173) underwent a formal assessment of their knowledge of pharmacoeconomics during their BPharm programme. Table 3 shows that most students (88.0%) recalled being taught pharmacoeconomics through lectures. The number of hours allocated to pharmacoeconomics in students’ timetables ranged from 0.1 to 40 h (mean = 4.4; SD = 4.52). Of the 22 additional comments at the end of the survey, five students (22.7%) said that “pharmacoeconomics should be a subject/module/course on its own”.

### 3.4. Understanding of Pharmacoeconomic Concepts

Table 4 demonstrates that, of the 172 students indicating their level of understanding of pharmacoeconomic concepts, 40 (23.3%) signified an overall poor understanding of these concepts, whereas 83 (48.3%) had a fair understanding. Only over a quarter of students signified an overall good understanding (45/172), and 66.2% were able to correctly answer questions regarding the scope of pharmacoeconomics (104/157). However, 37.0% of students (57/154) wrongly indicated that “pharmacoeconomics calculates the costs of medicines and treatments only”. Only 23.4% of students (33/138) provided correct answers to each type of analysis, namely, cost-minimization analysis, cost-effectiveness analysis, cost-utility analysis, and cost-benefit analysis.

### 3.5. Relevance of Pharmacoeconomics in Practice

Overall, the majority of students (87.5%; 140/160) perceived the application of pharmacoeconomics in South African medicines management as “relevant” (Table 5). Of the 22 additional comments, most students (88.8%) felt that applying pharmacoeconomics in practice was an essential skill that pharmacists should possess, with 84.0% indicating that pharmacists should be responsible for performing pharmacoeconomic evaluations in practice.

### 3.6. Preparedness for Application of Pharmacoeconomics in Practice

Only over a third of students (38.1%; 64/168) felt that their undergraduate exposure to pharmacoeconomics was insufficient to understand basic principles (Table 6). Nearly half of the students (54.2%; 91/168) perceived pharmacoeconomics as “interesting” and “enjoyable”, with 47.0% (79/168) who felt not adequately prepared to apply pharmacoeconomics in practice. Less than half (45.7%; 75/164) of the students thought of themselves as competent to perform basic pharmacoeconomic analyses.

### 3.7. Sensitivity Analysis

Based on the universities’ response rates (see Table 1), Universities A and C were grouped into “Low response rate“, Universities B and E into “Medium response rate“, and University D labelled as “High response rate”. The mean scores for understanding of basic pharmacoeconomic concepts (*p* = 0.006) and advanced pharmacoeconomic concepts (*p* < 0.001) were statistically significantly different between the three groups (see Table 7). However, post-hoc analysis showed no significant difference between the high- and low-response rate groups for basic (*p* = 0.991) and advanced (*p* = 0.774) understanding of pharmacoeconomic analyses. In terms of “Preparedness to apply pharmacoeconomics in practice,” there was no statistically significant difference between the three groups for both being able to interpret the results of pharmacoeconomic analyses for decision-making (*p* = 0.810) and being adequately prepared to apply pharmacoeconomic concepts in practice to conduct the analyses (*p* = 0.792) (see Table 8).

### 3.8. Future Education in Pharmacoeconomics

The vast majority of students (93.8%; 152/162) believed that future education regarding pharmacoeconomic studies and their application was essential to their role as pharmacists, while only five (3.1%) perceived further education as “not necessary”. Similarly, most surveyed students (84.4%; 135/160) would have wanted more education on pharmacoeconomics during their BPharm training. Among the additional comments provided, 27.2% (6/22) of the students thought that undergraduate pharmacoeconomics tuition should be increased.

The majority of students (86.7%; 137/158) wanted to acquire further pharmacoeconomics knowledge, of whom two thirds (66.4%; 91/137) wanted to acquire further knowledge through continuous professional development (CPD) programmes, 36 (26.3%) through self-study, and 40 (29.2%) through postgraduate studies. Among the 22 additional comments at the end of the survey, three students (13.6%) indicated that they would like to acquire more knowledge regarding pharmacoeconomics.

## 4. Discussion

The overall response rate of 38.1% is seen as acceptable for voluntary questionnaire surveys, and similar to other published studies in this area [44,45,47,55,56].

The study results principally highlighted two key issues for the future, which are the most important outcomes of this study. First, few pharmacy students had an overall good understanding of pharmacoeconomic concepts. However, only over half of those surveyed felt they received enough teaching exposure to pharmacoeconomics to understand the basic principles and concepts during their university training. However, the majority wanted to receive more undergraduate training and tuition. Second, less than half of the students participating in the survey felt competent to perform basic pharmacoeconomic analyses, with more students considering themselves as “not prepared” to conduct these studies compared with those who feel prepared.

Most students in our study underwent education regarding pharmacoeconomics in their fourth (final) year. This is consistent with the study by Catić and Skrbo in Bosnia, in which most pharmacy students were taught pharmacoeconomics in their fifth (final) year [16]. We assume that students nearing the end of their BPharm studies have the necessary fundamental medicine-related knowledge to fully understand pharmacoeconomics and its application. However, this is not always the case.

According to most students in this study, pharmacoeconomics was a mandatory subject/module/course, which concurs with the findings from similar studies [3,16,19,20,57,58,59,60]. However, the mean hours spent teaching pharmacoeconomics in South Africa varied significantly compared with similar studies outside of South Africa [3,17,19,47,50]. This might help in explaining why more than a third of students in our study felt they did not receive sufficient undergraduate exposure to pharmacoeconomics and its principles. This may have adversely affected their understanding of pharmacoeconomic concepts, competence to perform basic analyses, and preparedness to conduct pharmacoeconomic analyses in practice, which urgently needs to be addressed [61].

Encouragingly, few students in our study had an overall poor understanding of pharmacoeconomic concepts or principles. This compares with other studies, where most respondents found these concepts unclear or difficult to understand or indicated that they were “not very familiar” or “slightly familiar” with these concepts [16,46,49,61]. At the same time, the number of students in our study with an overall good understanding of pharmacoeconomic concepts and their application in medicines management was similar to other studies [44,45,47,49]. When questioned about the scope of pharmacoeconomic analyses, more students in our study knew the correct answer to the question, compared with 39.0% of students in the study by Catić and Skrbo [16]. However, more students in our study incorrectly indicated that “pharmacoeconomics examines and calculates the costs of medicines and treatments only” compared with only 13.0% of the respondents recorded in the study by Catić and Skrbo [16].

Of concern, many students in our study incorrectly indicated that budget impact analysis formed part of the scope of pharmacoeconomics, while not being considered as a pharmacoeconomic analysis sub-type in reality. This is an issue to address as budget impact analyses are increasingly important in LMICs when assessing the possible role and value of new medicines [62,63,64,65]. This is balanced against the finding that the number of students correctly indicating that pharmacoeconomic studies compare different therapeutic interventions was appreciably higher than only 7.0% of students reported by Catić and Skrbo [16]. In general, more students in our study had a good knowledge of the scope of pharmacoeconomics compared with only 38.9% of students reported by Catić and Skrbo [16]. Nearly a quarter of students in our survey were also able to provide the correct measure to each of the basic pharmacoeconomic analyses concurrent with similar studies [45,46,49]. The level of pharmacoeconomics understanding among the students in our study, especially regarding the scope of pharmacoeconomics, is an important factor to consider going forward. Any healthcare professional tasked with even the most basic pharmacoeconomic analyses would have to know what inputs and outcomes they are measuring when performing these evaluations in practice. Evidently, the level of students’ understanding of pharmacoeconomics is a fundamental cornerstone of their ability to perform pharmacoeconomic analyses successfully. Consequently, education offers a golden opportunity to improve their knowledge and understanding of pharmacoeconomics. The importance of addressing this gap in BPharm students’ knowledge and understanding is supported by earlier studies highlighting that the education of healthcare professionals regarding pharmacoeconomics contributes to the financial sustainability of healthcare systems [3,43,46,48].

Moreover, encouragingly, an appreciable portion of students in our study thought that applying pharmacoeconomics was an essential skill that South African pharmacists should possess. This concurs with the findings of studies conducted amongst different student cohorts in India, Japan, and South Africa [3,43,45,49]. Pharmacists, with their unique knowledge of medicines and key aspects of it, including their acquisition costs, can effectively contribute to any conservation regarding health budgets, which is important in enhancing equal access to pharmaceutical care, especially in developing countries such as South Africa [3,15,66,67]. Consequently, it was encouraging that most students in our study agreed that the application of pharmacoeconomics would benefit the South African healthcare system. This finding strongly correlates with Tahashildar et al. in India [45]. Moreover, how students thought of instances in which pharmacoeconomic evaluations could be used in South Africa was congruent with the findings by Catić and Skrbo from Bosnia and those of Modiba from South Africa [16,43].

Furthermore, encouragingly, most of the surveyed students intended to complete their internship in South African public sector institutions, challenged with significant workforce shortages as well as medicine shortages, requiring pro-active ways to deal with this without seriously impeding on patient care [9,26,68,69]. This is important for the future as South Africa implements UHC with ever-increasing demands on scarce resources, starkly contrasting the findings of the study by Armstrong et al. [3,26,49,70].

Only over a third of the students in our current study felt that they needed to receive more pharmacoeconomics exposure in their training. In addition, the majority of those surveyed would have wanted more pharmacoeconomics training at an undergraduate level. This is similar to a 2002 European Pharmaceutical Student Association survey involving 22 European countries, where only over half of the students surveyed (56%) indicated that the level of pharmacoeconomics education they received during their education was poor [3]. This needs addressing in the future, especially as only 19.0% of pharmacy students enrolled at the Lebanese American University School of Pharmacy believed that the number of hours spent preparing them to analyse pharmacoeconomic research was inadequate [71].

Most healthcare professionals and postgraduate medical students participating in the study by Tahashildar et al. did not feel comfortable in conducting pharmacoeconomic analyses. This was despite having undergone a formal assessment of their pharmacoeconomics knowledge [45]. There were similar findings in the study by Umair Khan et al. [48]. Both studies concur with our study, where only a limited number of students surveyed felt prepared and competent to perform basic pharmacoeconomic analyses in practice [45,48]. This study finding is emphasised by Kolassa, who suggested that pharmacy curricula did not adequately prepare students [72].

Our study also showed that half of the South African students wanted to obtain additional pharmacoeconomics knowledge through CPD programmes post qualification. This compares to approximately three-quarters of postgraduate students surveyed in the study by Jayasree et al. [44]. More students in our study wanted to acquire further pharmacoeconomics knowledge through self-study, which contrasts with 11.0% in the study by Jayasree et al. [44]. However, the small number of students in our study wanting to acquire pharmacoeconomics knowledge through postgraduate studies corresponded with findings by Catić and Skrbo, but was in contrast with the 55.0% of respondents in the study by Jayasree et al., who believed that pharmacoeconomics should be included in postgraduate studies [16,44].

We are aware of the limitations with this study. Firstly, four of the nine universities in the country offering the BPharm programme did not participate in the study during the study period as a number of them were unable to provide ethical approval in time. Secondly, two of the five participating universities had a lower response rate compared to the other universities. However, the overall rates are similar to other published studies in this area [44,45,47,55,56]. In this regard, a strength of this study was that it was conducted in the final semester of the BPharm study, assuming that students would have undergone the necessary education to participate. In addition, the study’ the future requirements for pharmacoeconomic teaching for student pharmacists across South Africa to be able to equip them for the future. Consequently, we feel that the overall response rate from 189 pertinent students is extremely helpful, with the findings seen as robust in providing direction for the future.

Despite the above, we recognise that the variation in response rates among the universities could have introduced a non-response bias, however, based on the sensitivity analysis results (see Table 7 and Table 8) it is unlikely that variation in the response rate could explain the variation in the study outcomes among the universities. The observed variation could possibly be explained by variations in the academic performance amongst the universities, such as staff to student ratio and emphasis on the teaching of pharmacoeconomics, emphasising standardisation of pharmacoeconomic curricula amongst universities in South Africa.

## 5. Conclusions

There is a recognised need to develop a pool of South African personnel who are able to conduct and evaluate pharmacoeconomic analyses as South Africa moves toward UHC. Consequently, it was encouraging to see that most BPharm students surveyed perceived pharmacoeconomics in South African medicines management as relevant to their future needs. In addition, a demand for further pharmacoeconomics education exists among the next generation of pharmacists. South African undergraduate pharmacy students appear to correlate with their international counterparts regarding the gap in their understanding and knowledge of pharmacoeconomic concepts and their preparedness for practical application, which needs addressing going forward.

Consequently, pharmacoeconomics education should remain in the South African BPharm programme curriculum; however, the current content requires expansion. Addressing this gap during South African undergraduate pharmacy education should increase students’ understanding of pharmacoeconomic concepts and their preparedness for applying these analyses in practice post qualification to benefit the South African healthcare system. The BPharm curriculum is currently under review and we will continue to monitor it in future research projects.

## Figures and Tables

**Table 1 pharmacy-11-00054-t001:** Response rate per university.

University	Number of Final Year Bachelor of Pharmacy Students, 2018	Response Rate	
		Number (*n*)	Percentage (%)
A	80	44	55.0
B	94	17	18.1
C	141	58	41.1
D	49	46	93.9
E	132	24	18.2
**Total**	**496**	**189**	**38.1**

Keys: A, C, D, E = Universities where students responded through paper-based questionnaires; B = University where students responded through online questionnaires; A–D = Universities from which data were collected in 2018; the number of final year Bachelor of Pharmacy students are given for 2018; Source: South African Pharmacy Council (SAPC), 16 February 2018; E = University from which data were collected in 2019; the number of final year Bachelor of Pharmacy students are given for 2019.

**Table 2 pharmacy-11-00054-t002:** Respondent demographics.

Student Demographics		Number (%)
**Gender** **(*n* = 189)**	Female	135 (71.4)
	Male	54 (28.6)
**Race** **(*n* = 187)**	Black African	140 (74.9)
	Coloured	7 (3.7)
	Indian	16 (8.6)
	White	24 (12.8)
**Age** **(*n* = 182)**	Younger than 22 years	12 (6.6)
	22–25 years	126 (69.2)
	Older than 25 years	44 (24.2)
**Degrees other** ** than BPharm** **(*n* = 13)**	Master of Medicine in Physiology	1 (7.7)
	Bachelor of Science	2 (15.4)
	Bachelor of Science (Honours) in Cellular Biology	1 (7.7)
	Bachelor of Science (Honours) in Microbiology	1 (7.7)
	Bachelor of Science in Biochemistry	1 (7.7)
	Bachelor of Science in Biochemistry and Physiology	1 (7.7)
	Bachelor of Science in Biological Science	1 (7.7)
	Bachelor of Science in Medicinal Science	1 (7.7)
	Bachelor of Science in Microbiology and Biochemistry	1 (7.7)
	Bachelor of Science in Molecular Life Sciences	3 (23.1)
**Intended sector ** ** of internship ** **(*n* = 179)**	Academic institution	19 (10.6)
	Community pharmacy	27 (15.1)
	Manufacturing pharmacy	10 (5.6)
	Private sector institutional pharmacy	14 (7.8)
	Public sector institutional pharmacy	88 (49.2)
	The respondent did not know	21 (11.7)

**Table 3 pharmacy-11-00054-t003:** Pharmacoeconomics education during the Bachelor of Pharmacy programme.

Pharmacoeconomics Education		Number (%)
**Subject/module/course under which pharmacoeconomics was offered (*n* = 164) ***		
	Community Pharmacy-Based Learning | Community Pharmacy Practice | Community Pharmacy	10 (6.1)
	Hospital Community Pharmacy | Hospital Pharmacy and Community Pharmacy | Hospital Pharmacy Practice and Community Pharmacy Practice	4 (2.4)
	Hospital Management | Hospital Pharmacy | Hospital Pharmacy Management | Hospital Pharmacy Practice | Hospital Pharmacy Practice-Based Learning | Hospital-Based Learning Module | Hospital-Based Pharmacy	62 (37.8)
	Natural Products and Evidence	1 (0.6)
	Pharmaceutical Care	1 (0.6)
	Pharmaceutical Logistics, Economics and Management | Pharmaceutical Logistics | Pharmaceutical Management	51 (31.1)
	Pharmacology	2 (1.2)
	Pharmacy and the Professional Environment | Pharmacy Practice and the Professional Environment | Pharmacy Practice | Philosophy of Pharmacy Practice	19 (11.6)
	Pharmacy People and Systems	8 (4.9)
	Pharmacy Research Project | Pharmacy Research | Research Methodology | Research Module	6 (3.7)
	Specialised Pharmacy and Pharmacoeconomics | Specialised Pharmacy	3 (1.2)
	Third Year level	68 (38.2)
	Fourth Year level	74 (41.6)
	Both third and fourth-year levels	36 (20.2)
	Elective subject/module/course	4 (2.3)
	Mandatory subject/module/course	158 (89.3)
	Inclusion of pharmacoeconomics other than mandatory or elective subject/module/course	2 (1.1)
	The respondent did not know	13 (7.3)
**Forms of pharmacoeconomics knowledge assessment during BPharm programme (*n* = 128) ***		
	Assignments	15 (11.7)
	Calculations	4 (3.1)
	Exams	81 (63.3)
	Multiple choice questions	5 (3.9)
	Oral exam	5 (3.9)
	Portfolios	1 (0.8)
	Practical work | Experimental learning | PTC-meeting attendance | Taking part in analyses	4 (3.1)
	Presentations	3 (2.3)
	Scenario-based questions | Case studies | Application questions | In-depth questions	6 (4.7)
	Summative and formative assessments | Assessments | Final assessments | Module assessments	8 (6.3)
	Tests (Semester tests | Class tests)	70 (54.7)
	Workshops	1 (0.8)
**Medium of pharmacoeconomics teaching during BPharm programme (*n* = 175) ***		
	Lectures	154 (88.0)
	Practical work	45 (25.7)
	Tutorials	37 (21.1)
	Workshops	45 (25.7)
	Another medium of teaching pharmacoeconomics	4 (2.3)
**Hours of pharmacoeconomics training received during BPharm programme (*n* = 100)**		
	Less than 1 h	20 (20.0)
	1–4 h	46 (46.0)
	5–7 h	14 (14.0)
	More than 7 h	20 (20.0)

* Respondents were allowed more than one answer to the question; PTC: Pharmaceutical and Therapeutics Committee.

**Table 4 pharmacy-11-00054-t004:** Understanding of pharmacoeconomic concepts and their application by South African final year Bachelor of Pharmacy students.

Understanding of Pharmacoeconomic Concepts and Their Application		Number (%)			
		Poor	Fair	Good	NA
**Level of ** **understanding**	Basic pharmacoeconomic concepts (*n* = 172)	26 (14.9)	92 (53.6)	51 (29.7)	3 (1.8)
	Advanced pharmacoeconomic concepts (*n* = 172)	53 (30.8)	74 (43.0)	40 (23.8)	5 (2.9)
				**Correct answer**	
**What the scope ** **of pharmacoeconomics entails**	Evaluates and measures costs and benefits of drug therapy (*n* = 162)			154 (95.1)	
	Involves a combination of health economics and clinical outcomes (*n* = 156)			145 (92.9)	
	Compares different pharmaceutical interventions to each other (*n* = 154)			125 (81.2)	
	Examines and calculates costs of medicines and treatments only (*n* = 154)			89 (57.8)	
	Measures the impact of medicine-related costs on medicine budgets (*n* = 156)			5 (3.2)	
				**Correct answer**	
**How outcomes for ** **pharmacoeconomic ** **analyses are measured**	CBA outcomes may be similar or different units and always expressed in monetary units (*n* = 139)			28 (20.1)	
	CEA outcomes are measured in similar natural health units across therapies (*n* = 139)			30 (21.6)	
	CMA outcomes are measured by assuming that health benefits are equivalent and can take any form (*n* = 138)			26 (18.8)	
	CUA outcomes value health benefits across therapies in similar units, depending on individual preference (*n* = 136)			45 (33.1)	

NA: Not applicable; CBA: Cost-benefit analysis; CEA: Cost-effectiveness analysis; CMA: Cost-minimisation analysis; CUA: Cost-utility analysis.

**Table 5 pharmacy-11-00054-t005:** Perceptions of final year Bachelor of Pharmacy students regarding the relevance of pharmacoeconomics in South African medicines management.

	Number (%)				
**The relevance of pharmacoeconomics ** ** in South African medicines management**	**Irrelevant**		**Neutral** **No opinion**	**Relevant**	
	**Strongly** **disagree**	**Disagree**		**Agree**	**Strongly agree**
Improves medicine-related decisions in South African healthcare system [*n* = 159]	7 (4.4)		8 (5.0)	144 (90.6)	
	5 (3.1)	2 (1.3)		69 (43.4)	75 (47.2)
Ensures optimal use of medicine budgets across the South African public health sector [*n* = 159]	11 (6.9)		6 (3.8)	142 (89.3)	
	7 (4.4)	4 (2.5)		79 (49.7)	63 (39.6)
Should form an integral part of the South African National Health Insurance system [*n* = 159]	9 (5.7)		12 (7.5)	138 (86.8)	
	6 (3.8)	3 (1.9)		67 (42.1)	71 (44.7)
Will improve access to medicines [*n* = 160]	7 (4.4)		17 (10.6)	136 (85.0)	
	5 (3.1)	2 (1.3)		83 (51.9)	53 (33.1)
**Instances of pharmacoeconomic analyses ** **application in South Africa**	**Not used**		**Neutral** **No opinion**	**Used**	
	**Strongly** **disagree**	**Disagree**		**Agree**	**Strongly agree**
Pricing of medicines [*n* = 158]	8 (5.1)		14 (8.9)	136 (86.1)	
	3 (1.9)	5 (3.2)		89 (56.3)	47 (29.7)
Planning of production and sales of medicines [*n* = 158]	8 (5.1)		16 (10.1)	134 (84.8)	
	4 (2.5)	4 (2.5)		100 (63.3)	34 (21.5)
Inclusion of medicines in medicine formularies (e.g., EML, medical aid formularies, STGs) [*n* = 158]	11 (7.0)		25 (15.8)	122 (77.2)	
	6 (3.8)	9 (5.7)		70 (44.3)	52 (32.9)
Clinical decision-making at the individual patient level, in the case of medicine not included in EML, medical aid formularies, STGs [*n* = 158]	12 (7.6)		29 (18.4)	117 (74.1)	
	4 (2.5)	8 (5.1)		70 (44.3)	47 (29.7)
Registration of new medicines with the South African Health Products Regulatory Authority [*n* = 158]	15 (9.5)		30 (19.0)	113 (71.5)	
	6 (3.8)	5 (3.2)		75 (47.5)	38 (24.1)
	**Not important**		**Neutral** **No opinion**	**Important**	
	**Strongly** **disagree**	**Disagree**		**Agree**	**Strongly agree**
Pharmacoeconomics is an important skill that South African pharmacists should possess [*n* = 160]	7 (4.4)		11 (6.9)	142 (88.8)	
	6 (3.8)	1 (0.6)		69 (43.1)	73 (45.6)
				**Number (%)**	
**South African health sector** ** to which pharmacoeconomic** ** application is relevant [*n* = 159]**	The private sector only (e.g., medical aid formularies)			6 (3.8)	
	Public sector only (e.g., EML, STGs)			11 (6.9)	
	Both private and public sector			134 (84.3)	
	The respondent did not know			8 (5.0)	
**Professions to perform** ** pharmacoeconomic analyses** ** in practice [*n* = 163]**	Health economists			140 (85.9)	
	Pharmacists			137 (84.0)	
	Medical practitioners			71 (43.6)	
	Economists			64 (39.3)	
	Accountants			52 (31.9)	
	Nursing practitioners			42 (25.8)	
	Epidemiologists			33 (20.2)	
	Mathematical modellers			20 (12.3)	
	Demographers			16 (9.8)	
	People with mathematical background			15 (9.2)	

EML: Essential medicines list; STGs: Standard treatment guidelines.

**Table 6 pharmacy-11-00054-t006:** Preparedness of South African final year Bachelor of Pharmacy students for application of pharmacoeconomics in medicines management.

		Number (%)				
**Opinions on pharmacoeconomics**		**Negative**		**Neutral** **No opinion**	**Positive**	
		**Strongly** **disagree**	**Disagree**		**Agree**	**Strongly agree**
Exposure to pharmacoeconomics during the BPharm programme was sufficient to understand basic principles and concepts [*n* = 168]		64 (38.1)		13 (7.7)	91 (54.2)	
		16 (9.5)	48 (28.6)		75 (44.6)	16 (9.5)
Pharmacoeconomics is interesting and enjoyable [*n* = 168]		40 (23.8)		37 (22.0)	91 (54.2)	
		11 (6.5)	29 (17.3)		65 (38.7)	26 (15.5)
**Preparedness to apply pharmacoeconomics in practice**		**Unprepared**		**Neutral** **No opinion**	**Prepared**	
		**Strongly ** **disagree**	**Disagree**		**Agree**	**Strongly agree**
Knows where to find more information on pharmacoeconomic concepts [*n* = 168]		48 (28.6)		25 (14.9)	95 (56.5)	
		11 (6.5)	37 (22.1)		77 (45.8)	18 (10.7)
Can interpret results of pharmacoeconomic analyses for decision-making [*n* = 165]		59 (35.8)		29 (17.6)	77 (46.7)	
		13 (7.9)	46 (27.9)		71 (43.0)	6 (3.6)
Adequately prepared to apply pharmacoeconomic concepts in practice to conduct analyses [*n* = 168]		79 (47.0)		31 (18.5)	58 (34.5)	
		15 (8.9)	64 (38.1)		50 (29.7)	8 (4.8)
		**Negative**		**Neutral** **No opinion**	**Positive**	
		**Very ** **incompetent**	**Incompetent**		**Competent**	**Very competent**
Competence in performing basic pharmacoeconomic analyses [*n* =164]		45 (27.4)		44 (26.5)	75 (45.7)	
		10 (6.1)	35 (21.5)		68 (41.7)	7 (4.3)
		**Never**		**Rarely**	**Often**	
**Expected frequency ** **of performing ** **pharmacoeconomic ** **analyses**	Cost-minimisation analysis [*n* = 164]	23 (14.0)		62 (37.8)	79 (48.2)	
	Cost-benefit analysis [*n* = 161]	19 (11.8)		64 (39.8)	78 (48.4)	
	Cost-effectiveness analysis [*n* = 163]	21 (12.9)		72 (44.2)	70 (42.9)	
	Cost-utility analysis [*n* = 163]	24 (14.7)		74 (45.4)	65 (39.9)	

**Table 7 pharmacy-11-00054-t007:** Mean level of understanding of pharmacoeconomic concepts distributed by university response rate.

Level of Understanding of Pharmacoeconomics		University D	Universities B and E	Universities A and C
		High Response Rate	Medium Response Rate	Low Response Rate
	** *n* **	45	95	31
**Basic pharmacoeconomic ** **concepts**	** Mean (SD)**	2.316 (0.5697)	2.036 (0.6123)	2.332 (0.4423)
	*** p*-value ***	0.006		
**Advanced pharmacoeconomic ** **concepts**	**Mean (SD)**	2.271 (0.4251)	1.796 (0.6135)	2.184 (0.4872)
	*** p*-value ***	<0.001		

* One-way ANOVA.

**Table 8 pharmacy-11-00054-t008:** Preparedness for the application of pharmacoeconomics in practice distributed by university response rate.

Preparedness to Apply Pharmacoeconomics in Practice		University D	Universities B and E	Universities A and C
		High Response Rate	Medium Response Rate	Low Response Rate
**Can interpret results of pharmacoeconomic analyses for** **decision-making**	** *n* **	44	93	28
	**Agree; No (%)**	22 (50.0)	40 (43.0)	15 (53.6)
	**Neutral; No (%)**	19 (43.2)	44 (47.3)	12 (42.9)
	**Disagree; No (%)**	3 (6.8)	9 (9.7)	1 (3.6)
	** *p* ** **-value ***	0.810		
**Adequately prepared to apply pharmacoeconomic concepts in practice to conduct analyses**	** *n* **	46	94	28
	**Agree; No (%)**	17 (37.0)	31 (33.0)	10 (35.7)
	**Neutral; No (%)**	26 (56.5)	55 (58.5)	14 (50.0)
	**Disagree; No (%)**	3 (6.5)	8 (8.5)	4 (14.3)
	** *p* ** **-value ***	0.792		

* Fisher Exact.

## Data Availability

Additional data are available upon reasonable request from the corresponding author.

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
