# Peer review of "Perceptions of and Preparedness for the Application of Pharmacoeconomics in Practice, among Final Year Bachelor of Pharmacy Students in South Africa: A National Cross-Sectional Study"

_pharmacy, 2023, doi:10.3390/pharmacy11020054_

Round 1
Reviewer 1 Report
This study is very important and necessary. The main problem with the study that I have is the very low response rate, i.e., 38.1% with only 5 of 9 universities represented in the study. This relates to a nonresponse bias of 61.8%. Therefore, the applicability of the results to be an assessment really limits the generalisability of the findings. For the 5 universities the response rate varied from 18.1% to 93.9%. It is this reviewers observation that numerous efforts should have been made to encourage nonrespondents to complete the assessment.
Author Response
Reviewer 1
(x) Extensive editing of English language and style required
Author comments: We have updated the paper with the help of one of the co-authors who is a native English speaker with over 450 publications in peer-reviewed Journals. We hope this is now acceptable
Comments
This study is very important and necessary.
Author comments: Thank you for this.
The main problem with the study that I have is the very low response rate, i.e.,38.1% with only 5 of 9 universities represented in the study. This relates to a nonresponse bias of 61.8%. Therefore, the applicability of the results to be an assessment really limits the generalisability of the findings. For the 5 universities the response rate varied from 18.1% to 93.9%. It is this reviewer’s observation that numerous efforts should have been made to encourage nonrespondents to complete the assessment.
Author Comments: Thank you for this comment. As seen, this type of response is seen across studies of this type – which we now highlight in the limitations section at the end with appropriate references. It must be noted that the data collection period was also extended to two years, as authors tried to get as much response as possible from the various universities. We are confident though that the findings are of interest and show the current needs of the Pharmacy students going forward as South Africa strives towards UHC and the implications for future investment decisions with multiple competing demands for both infectious and non-infectious diseases – building on a number of recent studies in SA and beyond (now referenced). We hope this is acceptable to you.

Reviewer 2 Report
Line 24, (cur-riculum content), correct it please
Good flow of abstract, it is well designed and well-written
You mentioned in the introduction the following (Pharmacoeconomic submissions to the South African National Department of Health 55 (NDoH) Pricing Committee have taken place voluntarily and for selected medicines in the 56 private health sector [16]. ) Please clarify more and why they use this approach?
You used the word (However) in line 58 and line 61, and word (because) in different places at the same talk. This is too confusing! Please re-write them in simple way and in clear way
You mentioned the line 64-68 (in near future), it is unacceptable in scientific paper as we can not establish any evidence will happen in future. Is the near future is in practice yet? Please re-write it
Line 74-75 (re-write it please, draw attention to the structure please)
Line 87 (have still to be fully investigated [3].) fully investigated !!?? would you use another term appropriate to your topic please?
Line 92 (. However, this was not taken further.) .. why , you need to say the reason behind that.
Line 134-135 (five- 134 point Likert scale).. would you explain in discussion what the advantage and disadvantage of the scale, and what are the alternatives for more precision
Line 151-152-153 (the open ended questions and responses will be categorized, that means you have qualitative and quantitative - Mixed methodology) am I right? Please mention that in details
In general, this research give no enough benefits for the readers from overseas as this only deal with the heath care system in South Africa. Moreover, the subsidy of government to the medications will be based on the utility and successfully clinical trials to the nominated medications.
Author Response
Comment: Line 24, (cur-riculum content), correct it please.
Author comment: Thank you – now also corrected others.
Comment: Good flow of abstract, it is well designed and well-written.
Author comment: Thank you. We have also made minor changes to further improve the flow and hope this is now acceptable.
Comment: You mentioned in the introduction the following (Pharmacoeconomic submissions to the South African National Department of Health 55 (NDoH) Pricing Committee have taken place voluntarily and for selected medicines in the 56 private health sector [16]. ) Please clarify more and why they use this approach?
Author Comment: Thank you – we have appreciably upgraded this section with additional references including references of recent pharmacoeconomic analyses in South Africa, and hope this is now acceptable.
Comment: You used the word (However) in line 58 and line 61, and word (because) in different places at the same talk. This is too confusing! Please re-write them in simple way and in clear way
Author comment: Thank you – now updated. We hope this is now OK.
Comment: You mentioned the line 64-68 (in near future), it is unacceptable in scientific paper as we can not establish any evidence will happen in future. Is the near future is in practice yet? Please re-write it
Author comment: Thank you – we believe this will be the case based on our many interactions with key personnel in the MoH in South Africa – so have slightly modified this. We hope this is now OK.
Comment: Line 74-75 (re-write it please, draw attention to the structure please)
Author comment: Thank you – now re-written. We hope this is now acceptable.
Comment: Line 87 (have still to be fully investigated [3].) fully investigated !!?? would you use another term appropriate to your topic please?
Author comment: Thank you – now updated.
Comment: Line 92 (. However, this was not taken further.) .. why , you need to say the reason behind that.
Author comment: Thank you. We have now upgraded this section and hope this is now OK.
Comment: Line 134-135 (five- 134 point Likert scale).. would you explain in discussion what the advantage and disadvantage of the scale, and what are the alternatives for more precision
Author comment: Thank you for this. We have now expanded this section to state that the Likert scale is widely accepted Internationally for qualitative projects of this nature, and included references to back this up. We hope this is now acceptable.
Comment: Line 151-152-153 (the open ended questions and responses will be categorized, that means you have qualitative and quantitative - Mixed methodology) am I right? Please mention that in details
Author comment: Thank you for this comment. Unfortunately this type of analysis is called a descriptive cross-sectional study in the literature – hope this is OK.
Comment: In general, this research give no enough benefits for the readers from overseas as this only deal with the heath care system in South Africa. Moreover, the subsidy of government to the medications will be based on the utility and successfully clinical trials to the nominated medications.
Author comment: Thank you for this comment. We have appreciably upgraded the paper to include additional comments and papers/ references from other countries to enhance the utility of this paper espeically among countries seeking UHC. We hope this is now acceptable.

Round 2
Reviewer 1 Report
Low response rate necessitates rejection of this manuscript.
Author Response
Dear Reviewer
Thank you for reviewing our submission. We, however, believe that the response rate to this descriptive survey is significant and representative of the SA pharmacy education sector population. We did all that was possible to obtain ethical approval from other universities. we believe the survey results are still important enough to be used in the academic sector to inform the educational approach.
Reviewer 2 Report
I revised all the answers from the authors, and I feel this paper ready for publication
Author Response
Dear Reviewer
Thank you for accepting our submission
